# Implicit Associations between Adverbs of Place and Actions in the Physical and Digital Space

**DOI:** 10.3390/brainsci11111523

**Published:** 2021-11-17

**Authors:** Laila Craighero, Maddalena Marini

**Affiliations:** 1Department of Neuroscience and Rehabilitation, University of Ferrara, 44121 Ferrara, Italy; 2Center for Translational Neurophysiology, Istituto Italiano di Tecnologia, 44121 Ferrara, Italy; Maddalena.Marini@iit.it

**Keywords:** spatial cognition, Implicit Association Test, actions functions, digital content generation, digital content consumption, embodied cognition, adverb of space

## Abstract

Neuropsychological, behavioral, and neurophysiological evidence indicates that the coding of space as near and far depends on the involvement of different neuronal circuits. These circuits are recruited on the basis of functional parameters, not of metrical ones, reflecting a general distinction of human behavior, which alternatively attributes to the individual the role of agent or observer. Although much research in cognitive psychology was devoted to demonstrating that language and concepts are rooted in the sensorimotor system, no study has investigated the presence of implicit associations between different adverbs of place (*far* vs. *near*) and actions with different functional characteristics. Using a series of Implicit Association Test (IAT) experiments, we tested this possibility for both actions performed in physical space (*grasp* vs. *look at*) and those performed when using digital technology (*content* *generation* vs. *content* *consumption*). For both the physical and digital environments, the results showed an association between the adverb *near* and actions related to the role of agent, and between the adverb *far* and actions related to the role of observer. Present findings are the first experimental evidence of an implicit association between different adverbs of place and different actions and of the fact that adverbs of place also apply to the digital environment.

## 1. Introduction

The notion of egocentric distance refers to the space between an observer and a reference. All human languages contain special grammatical terms that serve to distinguish between different sectors of space, using the speaker or addressee as a frame of reference. Crosslinguistic research suggests that a large proportion of languages in the world make a fundamental binary distinction between terms that refer to something that is a short distance away and terms that refer to something that is a great distance away [1]. Among these terms, the adverbs of place serve to specify the place of action, the position of a person or an object in space, and the distance of a person or object from the speaker or listener. In particular, *far* is who or what is at a great distance, and *near* is who or what is close to where one is.

The discovery of the neurophysiological basis of the distinction of physical space into near and far has not been trivial. Indeed, at a level of physiology of vision, no index allows distinguishing in a binary way “near” from “far”. It is only possible to have visual information about which object is “closest”. Information on the distance between the observer and static objects is based on cues provided by the coordinated use of the two eyes, mainly stereopsis and eye convergence, which compare the parameters coming from different objects simultaneously present [2]. The situation is further complicated, as a series of studies showed that the metrics in perception are influenced by many different variables, such as the energetic costs associated with performing distance-relevant actions [3,4], the observer’s purposes [4], and the behavioral abilities of the observer’s body [5]. Specifically, targets at a distance looked farther when carrying a heavy load compared with carrying no load or a lighter load [3,4], but only if the intention was to throw the load (e.g., a ball) to it, not if participants intended to walk to it [4]. Furthermore, targets located between the boundaries of what cannot be reached without a tool but can be reached with one appear closer when the tool was used [5].

Neuropsychological studies investigating neglect patients [6,7] have shed some light on what is special about touching an object, with or without the help of a tool, to change the perception of distance. Neglect patients tend to ignore the left side of their visual field. For example, when they are asked to bisect a line, they bisect only the right half, resulting in responses far to the right of the true center. Some patients show neglect only for near lines and not for far lines [8], whereas other patients show neglect only for far lines [9]. This double dissociation constitutes strong evidence that the brain contains separate neural circuits for near or peripersonal space, on the one hand, and far or extrapersonal space, on the other. For patients who have shown neglect only in near space, it has been found that bisecting a distant line using a stick causes the error to occur. However, when patients are asked to use a laser pointer to bisect a distant line, the far-space error is not present [6,7]. Therefore, the use of the stick induces the appearance of a symptom in the far space that is normally present in the near space only, indicating that the far space has been remapped as near space as a result of the use of the stick. This remapping, however, does not occur when a laser pointer is used. Thus, these results suggest that the use of the stick and the pointer may involve different neuronal circuits. The stick may require the activation of the damaged circuit, responsible for the presence of peripersonal neglect. The pointer, on the other hand, may use a healthy circuit able to allow a perfect performance.

The main difference between the stick and the laser pointer is functional. With the stick, you reach the target, and you can exert such force that you can break the target, rotate it, push it, or make it feel touched in the case of an animated being. In the latter case, you can communicate something to the target, you can catch his attention, you can push him in a certain direction, stroke him with a light touch, or punish him with a heavy touch. The effects on the target are exactly those that would be produced by the use of the arm/hand. With the laser pointer, in contrast, you can point to something very precisely, but you are not able to physically modify the target. You can neither break it, nor move it, nor draw his attention if he is not looking at you. The laser pointer behaves in a similar way to the eyes; it has the same ability to focus on an extremely precise point of space from which we can obtain the information that is interesting at that moment. Several studies on monkeys, patients, and normal participants also support the view that tool use can modulate peripersonal space [10,11,12,13,14,15] and showed that this modulation has to be attributed to tool functionality, i.e., the possibility to interact and modify the state of the target, and not to other confounding variables, such as proprioceptive feedback, or visual appearance [16]. Indeed, the division of the space based on the potential actions that can be performed in it is reflected also by the organization of the sensorimotor system. Separate networks are devoted to the movement programming of the eyes [17,18,19] and to the encoding of reaching movements of the head, arm, and hand [20]. Therefore, the reachable peripersonal space and the unreachable extrapersonal space are anatomically and functionally represented in segregated circuits [21].

Taken together, the neuropsychological, behavioral, and neurophysiological evidence suggests that the binary cognitive/linguistic distinction of space into near and far is not defined by metrical parameters but by functional ones, that is, it depends on the possibility to voluntarily act on the target to modify its physical state. This position is in agreement with embodied cognition theories [22,23], according to which language and concepts are grounded in the sensorimotor system, given the presence of a deep interconnection between cognition, perception, and action. At present, there is very little experimental evidence that indicates a link between proximal and distal linguistic descriptors and the activation of the sensorimotor system. Specifically, it has been shown that, during a task to reach an object, the automatic reading of spatial adverbs (*far* vs. *near*) that are inconsistent with the real position of the object influences the kinematics of the reaching movement [24]. More interestingly, a study indicated that when participants were asked to name (for example, “this red triangle”; “that red triangle”) and to point with their hand or a tool to objects at different distances, the use of a stick led to greater use of the proximal demonstratives for objects placed at greater distances [25]. This latter result is in agreement with the hypothesis that the use of spatial demonstratives reflects a distinction between near and far space based on the actual possibility for the individual to act in that space at that moment.

However, as far as we know, there is no experimental evidence that indicates an association at a cognitive level between spatial adverbs and actions with different functional characteristics, in a context that does not involve the participant in the execution of an action. The first objective of this study was to fill this gap by studying the presence of implicit associations between adverbs of space and labels referring to different actions performed in the physical environment.

In addition, today, many of our activities are carried out using smartphones, and the COVID-19 pandemic has helped to increase the imbalance between actions in the physical and digital space. The digital space is an information space designed as a network of more or less static addressable objects, where information is perceived, stored, and retrieved, and where an individual can interact with others. Obviously, digital space is not constrained by metrical parameters, and we access to it by clicking the app icons which are present on the screen. Thus, digital space may be coded as peripersonal or extrapersonal according to the same rules that apply in the physical space, which is peripersonal when objects can be acted upon and a clear interaction is present and extrapersonal when objects can be just perceived. If so, then the app icons may reveal the potential actions to be performed on the target, and consequently they may be associated with the near or far space. Indeed, as with physical space, also with digital space, our behaviors may be divided into perceiving or acting. The terms used to categorize these different online behaviors are, respectively, “content consumption” and “content generation” [26]. Content consumption refers to the act of reading, listening, viewing, and other ways of taking in various forms of digital media. Examples of apps that involve this type of activity are those that allow access to web browsers (e.g., Google, Firefox, Safari), to weather information (e.g., 1 Weather, Weather Now, AccuWeather), and news (e.g., Apple News, Google News, The Week, Flipboard). Content generation, instead, describes the various practices that result in any type of digital content, including text and voice messages, video files, photos, etc., to be shared with the digital community via blogs, email apps, and social media sites (e.g., Facebook, WhatsApp, Instagram, Twitter). The second objective of this work was, therefore, to study for the first time the presence of implicit associations between spatial adverbs and app icons that direct to online actions with different functional characteristics.

Specifically, we propose that in both physical and digital environments the binary cognitive/linguistic distinction of space into near and far reflects an encoding in an operational/functional format (i.e., the boundary between the two regions depends on the possibility to voluntarily act on the target to modify its physical state). We aimed to test whether, in the physical environment, the adverb of space *near* is associated with actions that require reaching the target (e.g., reaching and grasping the object), and the adverb of space *far* with actions in which reaching the target is not allowed (e.g., looking at an unreachable object) and, congruently, whether in the digital environment, the adverb of space *near* is associated with app icons directing to content generation actions (e.g., WhatsApp) and the adverb of space *far* with those directing to content consumption actions (e.g., Google).

To this end, we used a series of Implicit Association Test (IAT) [27,28] experiments. The IAT is a research tool based on reaction time recordings for indirectly measuring the strength of associations among categories. The task requires sorting of stimulus exemplars from four categories using only two response options, each of which is assigned to two of the four categories. The logic of the IAT is that this sorting task should be easier when the two categories that share the same response option are strongly associated than when they are weakly associated. In the twenty years since its initial publication, the IAT has been applied in a diverse array of disciplines including social psychology [29,30], cognitive psychology [31,32], clinical psychology [33,34], developmental psychology [35,36], neuroscience [37,38,39,40,41], market [42], and health research [34,43,44].

Specifically, in the present study, four experiments were conducted using the IAT: (i) in the physical environment, to investigate the implicit association between the adverbs of space *near* and *far* and the actions *grasp* and *look at* (Experiment 1), and (ii) in the digital environment, to investigate the implicit association between the adverbs of space *near* and *far* and apps directing to *content generation* and *content consumption* actions (Experiments 2–4). We expected an association between *near* + *grasp*/*far* + *look at*, in the physical environment, and between *near* + *content generation apps*/*far* + *content consumption apps*, in the digital one.

The data for each experiment are available at the link: https://osf.io/9g235/?view_only=f072f3bb7ea34def93ea3cb07b9f2e78 (accessed on: 15 November 2021)

## 2. Experiment 1

The aim of Experiment 1 was to test for the presence of an implicit association between the adverbs of space *near* and *far*, needed to respectively classify objects at a reachable distance and landscapes and pictures showing individuals performing a reach and *grasp* action or a *look at* action. These actions are characterized by different functional characteristics. The reach and grasp action changes the state of the target as it moves it, whereas the look at action does not. According to the evidence reviewed in the Introduction, the binary distinction of space into near and far is defined by functional parameters, that is, it depends on the possibility to voluntarily act on the target to modify its physical state. Therefore, we expected an association *near* + *grasp*/*far* + *look at*.

### 2.1. Materials and Methods

#### 2.1.1. Participants

For all the experiments reported in this article, the following indications apply. Participants were unaware of the purposes of the study. Procedures were in accordance with the guidelines of the Declaration of Helsinki. Data collection was carried out from March to December 2020. Participants were actively recruited by sending an invitation on Google Classroom to all students enrolled in the different teaching courses at the University of Ferrara. Interested students contacted the authors via email granting informed consent. Each interested student was emailed a private link to the Project Implicit research platform (https://implicit.harvard.edu) to participate in the experiment. Participants were asked not to run the experiment more than once and not to share the link with others. Information about gender and age, but not on identity, was requested on the research platform before starting the experiment. All data were collected online, and no information about participants’ identity (neither name nor email address) was recorded. The number of data present for each experiment always corresponded to the number of students who answered the call for that experiment. We assume that no participant has taken part in more than one experiment as no student has been sent more than one link.

Forty-three participants (*M* age = 22.53, *SD* age = 2.32, 58.1% women) volunteered to take part in Experiment 1.

#### 2.1.2. Stimuli and Procedure

Stimuli used in Experiment 1 consisted of 20 colored pictures, subdivided into four categories: (i) for the *near* category, five images showing objects at a reachable distance; (ii) for the *far* category, five images showing landscapes; (iii) for the *grasp* category, images portraying a man or a woman grasping five different objects; (iv) for the *look at* category, images portraying a man or a woman looking at the same objects, at the same distance, with a Plexiglass placed between the actor and the object.

The link opened a webpage where participants were first asked to enter their gender and age and then displayed instructions for performing the task. In particular, a double entry table was presented in which images and associated categories were shown (Appendix A in Appendix A), accompanied by the information relative to sorting categories and response options. In all experiments, the two answer options were the “E” key (leftmost) and the “I” key (rightmost) on the QWERTY keyboard. Instructions were as follows: “In the center of the screen, one at a time, you will be presented with the IMAGES you see below. You will have to press the button corresponding to the CATEGORY to which the image belongs. When the category to which the image belongs is written at the top LEFT, you will have to quickly press the E key with the index finger of your LEFT hand. When the category to which the image belongs is written at the top RIGHT, you will have to quickly press the I key with the index finger of your RIGHT hand.”

Stimuli appeared one at a time in the middle of the screen, together with the names of the contrasted categories on the left and right at the top of the screen. If the participant made an error, a red X appeared below the stimulus and the trial continued until the correct key was pressed.

The experimental session lasted about 10 min on average and consisted of 180 trials subdivided into seven blocks, following the standard task procedure described by Nosek, Greenwald, and Banaji [45]. In this procedure, some blocks are practice tasks to acquaint subjects with the stimulus materials and sorting rules (Blocks 1, 2, 5). Others are critical blocks, which involve simultaneous sorting of stimulus items representing four concepts/categories with two response options (Blocks 3, 4, 6, 7). Ease of sorting is indexed by the speed of responding, faster responding indicating stronger associations.

In the present experiment, in Block 1 (action categories, 20 trials), participants practiced sorting images belonging to the *grasp* category (i.e., individuals grasping objects) and images belonging to the *look at* category (i.e., individuals looking at the objects). They were asked to press the E key for *grasp* and the I key for *look at*. In Block 2 (distance categories, 20 trials), participants practiced sorting images belonging to the *near* category (i.e., objects at a reachable distance) and images belonging to the *far* category (i.e., landscapes). They pressed the E key for *near* and the I key for *far*. In Blocks 3 (20 trials) and 4 (40 trials), participants categorized images belonging either to the action and distance categories (combined blocks, action + distance; congruent blocks). They pressed the E key for *grasp* and *near* and the I key for *look at* and *far*. In Block 5 (reverse action categories, 20 trials), participants practiced sorting images belonging to the action categories with the reverse key mapping from Block 1, i.e., *grasp* with the I key and *look at* with the E key. In Blocks 6 (20 trials) and 7 (40 trials), participants sorted images from all four categories with the opposite key pairings from Blocks 3 and 4 (combined blocks, reverse action + distance; incongruent blocks), i.e., *look at* and *near* with the E key and *grasp* and *far* with the I key (Figure 1).

The order of the action categories blocks (Blocks 1 and 5) and the combined blocks (Blocks 3 + 4 and Blocks 6 + 7) was counterbalanced across subjects. That is, half of the participants (Order 1) administered the blocks in the order just outlined, whereas the other half (Order 2) completed the task with Block 1, Block 3, and Block 4 switched with Block 5, Block 6, and Block 7 (Figure 2).

Response times and errors were collected online through the Project Implicit research platform (https://implicit.harvard.edu). Response time is the time from the onset of a single stimulus to the categorization of that stimulus.

### 2.2. Data Analysis

Data were analyzed according to the recommended IAT algorithm described by [46]. That is, we computed the *D* score for each participant. The *D* score [46] is a variation of Cohen’s *d* calculation of effect size used to measure the association strength between categories in the IAT. Research showed that the recommended IAT algorithm strongly outperforms, in terms of psychometric proprieties, the conventional procedures often used in cognitive and social psychology for RTs-paradigms, such as comparing latencies or errors in the combined blocks [27,46]. Indeed, this algorithm simultaneously considers both latencies and errors. That is, it uses a metric calibrated by each respondent’s latency variability and includes a latency penalty for errors.

Specifically, to calculate *D* score for each participant, we (a) removed responses faster than 350 ms and slower than 10,000 ms, (b) computed the mean of correct latencies for each combined block (Block 3, Block 4, Block 6, and Block 7), (c) computed one pooled standard deviation for all trials in Block 3 and Block 6 (*SD*_3–6_) and another for Block 4 and Block 7 (*SD*_4–7_), (d) replaced errors with the mean of the correct responses in that response block (computed in Step b) plus a 600 millisecond penalty, (e) averaged the resulting values for each of the four blocks (*M_Block_*_3_, *M_Block_*_4_, *M_Block_*_6_, *M_Block_*_7_), (f) computed the two mean differences (*M_Block_*_6_ − *M_Block_*_3_) and (*M_Block_*_7_ − *M_Block_*_4_), (g) divided each difference score by its associated pooled standard deviation, and (h) computed *D* as the equal-weight average of the two resulting ratios.
$$D=\frac{\left(\frac{M_{Block6}-M_{Block3}}{SD_{3-6}}\right)+\left(\frac{M_{Block7}-M_{Block4}}{SD_{7-4}}\right)}{2}$$

According to the present blocks sequence, *M_Block_*_6_ − *M_Block_*_3_ corresponded to
$$M_{Block6\left(Near–Look\;at/Far–Grasp\right)}-M_{Block3\left(Near–Grasp/Far–Look\;at\right)}$$
and *M_Block_*_7_ − *M_Block_*_4_ corresponded to
$$M_{Block7\left(Near-Look\;at/Far-Grasp\right)}-M_{Block4\left(Near-Grasp/Far-Look\;at\right)}$$

Therefore, a positive *D* score indicated an association *near + grasp/far + look at*.

The mean and the standard deviation of *D* scores of all participants were calculated.

To test the significance of the association revealed by the IAT, the mean *D* score was compared against zero (i.e., no association) using a one sample *t*-test.

### 2.3. Results

Results showed a positive and significant *D* score (*D_M_* = 0.31, *D_SD_* = 0.44, Cohen’s *d* = 0.70, *t*_(42)_ = 4.595, *p* < 0.001, 95% C.I. (0.17, 0.45)), indicating an association *near + grasp/far + look at*. That is, participants associated the adverb of space *near* more with the action category *grasp* and the adverb of space *far* more with the action category *look at* (Figure 6). The percent error was 5.64%.

### 2.4. Discussion

Results of Experiment 1 indicated the presence of implicit associations between adverbs of space and different actions performed in the physical environment. Specifically, they showed a significant association between the adverb *near* and the reach and *grasp* action and between the adverb *far* and the *look at* action.

Therefore, these results suggest that the use of proximal or distal adverbs of space is related to the potential possibility of acting to modify the objects present in that space.

## 3. Experiment 2

The aim was to use the IAT to study for the first time the presence of implicit associations between spatial adverbs and app icons that direct to online actions with different functional characteristics. The starting hypothesis was that not only the conceptual knowledge of the physical world is mapped within our sensorimotor system [22,23] but also that of the digital world. Digital experiences and the spaces in which they take place, even if they are sometimes called virtual, are quite real and have real, definite consequences [47]. Consequently, as well as the sensorimotor system characterizes the semantic content of concepts in terms of the way that we function with our bodies in the physical world [23], the sensorimotor system may characterize the concepts related to the digital world in terms of the way we act in it. Specifically, we expected that the distinction between adverbs of space *near* and *far* depends on the presence of digital actions that, respectively, involve or not a modification of the digital content.

Experiment 2 tested for the presence of an implicit semantic association between the adverbs *near* and *far*, needed to respectively classify objects at a reachable distance and landscapes, and *social* and *no social* app icons. Social app icons (e.g., Facebook, WhatsApp, Instagram, Twitter, etc.) address content generation actions, and no social app icons address content consumption actions (e.g., Google, Weather Now, The Week, etc.) We expected an association *near + social/far + no social*.

### 3.1. Materials and Methods

#### 3.1.1. Participants

Thirty-four participants (*M* age = 23.65, *SD* age = 4.59, 44.1% women) volunteered to take part in Experiment 2.

#### 3.1.2. Stimuli and Procedure

The stimuli used in Experiment 2 consisted of 20 colored pictures, subdivided into four categories: (i) for the *near* category, five images showing objects at a reachable distance; (ii) for the *far* category, five images showing landscapes; (iii) for the *social* category, images showing App icons addressing five social media sites (i.e., Instagram, WhatsApp, Twitter, Facebook, Snapchat); (iv) for the *no social* category, images showing App icons addressing five information media sites (i.e., Weather, iTunes, Google Maps, Chrome, Google) (Appendix A in Appendix A).

Procedure (Figure 3) and data analysis were the same as in Experiment 1.

According to the present blocks sequence, *M_Block_*_6_ − *M_Block_*_3_ corresponded to
$$M_{Block6\left(Near-No\;social/Far-Social\right)}-M_{Block3\left(Near-Social/Far-No\;social\right)}$$
and *M_Block_*_7_ − *M_Block_*_4_ corresponded to
$$M_{Block7\left(Near-No\;social/Far-Social\right)}-M_{Block4\left(Near-Social/Far-No\;social\right)}$$

Therefore, a positive *D* score indicated an association *near + social/far + no social*.

### 3.2. Results

Results showed a positive and significant *D* score (*D_M_* = 0.46, *D_SD_* = 0.48, Cohen’s *d* = 0.96, *t*_(33)_ = 5.609, *p* < 0.001, 95% C.I. (0.30, 0.63)), indicating an association of *near + social/far + no social*. That is, participants associated the adverb of space *near* more with the app category *social* and the adverb of space *far* more with the app category *no social* (Figure 6). The percent error was 5.64%.

### 3.3. Discussion

Results of Experiment 2 showed an implicit association between spatial adverbs and app icons that direct to online actions with different functional characteristics. Specifically, the difference between social and no social apps concerned the digital actions they addressed, that is, content generation and content consumption. Therefore, present results suggest that adverbs of space also apply to digital space.

However, some linguistics researchers criticize the idea that demonstratives are a particular class of spatial terms based on our bodily experience with concrete objects in space and propose that demonstratives are used primarily for social and interactive purposes [48]. This possibility questions our interpretation of the data. Indeed, it is possible that the association found between *near* and *social* and between *far* and *no social* depended on the label used to classify the apps, which explicitly referred to social activity. Unfortunately, no other labels could be found that would allow easy categorization of the icons.

## 4. Experiment 3

Experiment 3 was designed to test the hypothesis of an implicit association between spatial adverbs and different categories of app icons, without using a category label that refers to social activity. Since we could not find labels to categorize multiple apps, we decided to use only two apps, i.e., WhatsApp and a weather information app. WhatsApp allows users to send text messages and voice messages; make voice and video calls; and share images, documents, user locations, and other content. Undoubtedly, it is an app that requires content generation. On the contrary, using an app to check the weather requires content consumption. We used images of the most used weather forecasting apps in Italy (i.e., il Meteo, 3B Meteo, meteo.it).

### 4.1. Materials and Methods

#### 4.1.1. Participants

Twenty-six participants (*M* age = 22.42, *SD* age = 2.18, 53.8% women) volunteered to take part in Experiment 3.

#### 4.1.2. Stimuli and Procedure

The stimuli used in Experiment 3 consisted of 20 colored pictures, subdivided into four categories: (i) for the *near* category, five images showing objects at a reachable distance; (ii) for the *far* category, five images showing landscapes; (iii) for the *WhatsApp* category, images showing five different WhatsApp icons; (iv) for the *Weather* category, images showing five different weather forecasting Apps (Appendix A in Appendix A).

Procedure (Figure 4) and data analysis were the same as in Experiment 1.

According to the present blocks sequence, *M_Block_*_6_ − *M_Block_*_3_ corresponded to
$$M_{Block6\left(Near-Weather/Far-WhatsApp\right)}-M_{Block3\left(Near-WhatsApp/Far-Weather\right)}$$
and *M_Block_*_7_ − *M_Block_*_4_ corresponded to
$$M_{Block7\left(Near-Weather/Far-WhatsApp\right)}-M_{Block4\left(Near-WhatsApp/Far-Weather\right)}$$

Therefore, a positive *D* score indicated an association *near + WhatsApp/far + Weather*.

### 4.2. Results

Results showed a positive and significant *D* score (*D_M_* = 0.22, *D_SD_* = 0.48, Cohen’s *d* = 0.46, *t*_(25)_ = 2.384, *p* < 0.05, 95% C.I. (0.03, 0.42)), indicating an association of *near + WhatsApp/far + Weather.* That is, participants associated the adverb of space *near* more with the app icon *WhatsApp* and the adverb of space *far* more with the app icons *Weather* (Figure 6). The percent error was 5.10%.

### 4.3. Discussion

Results of Experiment 3 confirmed those of Experiment 2. They showed an implicit association between different adverbs of space and different categories of app icons. Specifically, *WhatsApp*, an app that requires content generation, was associated to the adverb *near*, while *Weather* forecasting apps, requiring content consumption, were associated to the adverb *far*.

However, even in this case, an alternative interpretation may question whether the results depended on the functional characteristics of the digital actions evoked by the app icons. Indeed, it is possible that the highlighted association depended on the presence of landscapes, a possible destination of a journey, and on the interest in weather that can jeopardize the decision to move.

## 5. Experiment 4

Experiment 4 aimed to test the hypothesis of an implicit association between spatial adverbs and categories of app icons related to content consumption and content generation, avoiding confounding variables related to category labels and the meaning of the stimuli. For this purpose, we selected WhatsApp and Google Chrome icons. Google Chrome is a web browser, that is, a software application for accessing information on the World Wide Web and, therefore, refers to content consumption.

### 5.1. Materials and Methods

#### 5.1.1. Participants

Forty-four participants (*M* age = 21.39, *SD* age = 3.25, 59.1% women) volunteered to take part in Experiment 4.

#### 5.1.2. Stimuli and Procedure

The stimuli used in Experiment 4 consisted of 20 colored pictures, subdivided into four categories: (i) for the *near* category, five images showing objects at a reachable distance; (ii) for the *far* category, five images showing landscapes; (iii) for the *WhatsApp* category, images showing five different WhatsApp icons; (iv) for the *Google* category, images showing five different Google icons (Appendix A in Appendix A).

Procedure (Figure 5) and data analysis were the same as in Experiment 1.

According to the present blocks sequence, *M_Block_*_6_ − *M_Block_*_3_ corresponded to
$$M_{Block6\left(Near-Google/Far-WhatsApp\right)}-M_{Block3\left(Near-WhatsApp/Far-Google\right)}$$
and *M_Block_*_7_ − *M_Block_*_4_ corresponded to
$$M_{Block7\left(Near-Google/Far-WhatsApp\right)}-M_{Block4\left(Near-WhatsApp/Far-Google\right)}$$

Therefore, a positive *D* score indicated an association *near + WhatsApp/far + Google*.

### 5.2. Results

Results showed a positive and significant *D* score (*D_M_* = 0.21, *D_SD_* = 0.40, Cohen’s *d* = 0.53, *t*_(43)_ = 3.549, *p* < 0.001, 95% C.I. (0.09, 0.33)), indicating an association of *near + WhatsApp/far + Google*. That is, participants associated the adverb of space *near* more with the app icon *WhatsApp* and the adverb of space *far* more with the app icon *Google* (Figure 6). The percent error was 4.94%.

### 5.3. Discussion

Results of Experiment 4 confirmed those of previous experiments, that is, we observed an implicit association between different adverbs of space, *near* and *far*, and the app icons of *WhatsApp* (digital content generation) and *Google* (digital content consumption), respectively.

## 6. General Discussion

The present study aimed to test the semantic association between adverbs of place and the concept of acting vs. perceiving. Specifically, we hypothesized that the use of adverbs *near* and *far* depends on the functional characteristics of the potentially performed actions in that space. This position stems from neuroscientific evidence pointing to separate neuronal circuits coding the near peripersonal space, that, is the area of space reachable by body parts, in which objects can be manipulated, and the far extrapersonal space, that is, the area of space out of reach, in which objects can be only perceived [49,50,51,52]. Consequently, we predicted a stronger association between *near* and actions able to act on the target and between *far* and actions devoted to the use or perception of target information. This binary distinction between actions with different functional characteristics is present not only in the physical space (e.g., to grasp, to move, to brake, etc. vs. to look at, to observe, to check, etc.) but also in the digital one. Digital behaviors are divided into content generation and consumption: the first produces new digital content to be published online, and the second is limited to the use of content already present on the network [26].

To this end, we used a series of IAT experiments [27,28] to investigate the associations of the category *near* (images showing objects at a reachable distance) and the category *far* (images showing landscapes), with actions characterized by different functional characteristics, performed in the physical (Experiment 1) and digital (Experiments 2–4) environment.

In Experiment 1, actions were grasp (images portraying individuals grasping objects) and look at (images portraying individuals looking at objects). Results confirmed the hypothesis, showing a stronger association between *near* and *grasp* and between *far* and *look at*. Note that in both grasp and look at stimuli, the objects were placed at the same distance from the agent. In the first case, the agent touched the object with the extended arm, in the second one, a transparent Plexiglas barrier was placed between the agent and the object, and the agent’s hand was resting on the table. This detail is important for several reasons. The first one is that it excludes that IAT results depended on a different distance between the agent and the object. A farther metric distance could easily be associated with the considerable distance from which you look at the landscapes used as stimuli for the far category. The second reason regards the necessity to carefully distinguish the encoding of peri- and extrapersonal space in a metric format (i.e., the boundary between the two regions depends on the distance from the agent’s body) from that in an operational/functional format (i.e., the boundary between the two regions depends on the workspace of the agent) [53]. We hypothesized that the implicit association between actions and adverbs of space reflects an encoding in a functional format. Indeed, the presence of the Plexiglas restricted the workspace, preventing the agent from reaching the object with his hands, without modifying its distance from the agent. Furthermore, it is known that, in the absence of barriers, the mere presence of a graspable object in the vicinity of an agent evokes in the observer the idea of an action towards the object [54,55].

In Experiments 2–4, actions were represented by different app icons addressing either content generation or content consumption behaviors. In all these experiments, results confirmed the hypothesis showing an association between *near* and *content generation* apps and between *far* and *content consumption* apps. While possible alternative explanations could apply for Experiments 2 and 3, we found no possible confounding variable linked to category labels, the meaning of the stimuli, or other that could explain the results of Experiment 4 in an alternative way. Specifically, present results suggest that adverbs of space also apply to digital space and that the distinction into near and far depends on the functional characteristics of digital actions addressed by the app icons. In future experiments, the usage of the digital app by the participants will also be taken into account. In fact, many people may be using platforms like Instagram or Twitter more for content consumption than for content generation. Current results already indicate a strong effect that would likely be reinforced if we could only consider participants using these apps for content generation. Furthermore, the next experimental steps will aim to generalize these results to contexts that do not contrast social activities with non-social activities. For example, we will test the hypothesis in a condition where content generation does not involve a social environment, e.g., a single-player game.

Regarding the physical space, several neuroscience findings showed the crucial role of action consequences in the coding of space, as extensively discussed in the Introduction (for a recent review, see [56]). However, to our best knowledge, results observed in the present study are the first experimental evidence of an implicit association between actions and adverbs of space at a cognitive level. Gallese et al. [23] proposed that conceptual knowledge is mapped within our sensorimotor system, which characterizes the semantic content of concepts in terms of the way that we function with our bodies in the world. This position, shared by others [57,58,59,60,61,62], is opposed to that considering concepts as represented outside of sensorimotor cortices [63,64,65]. Specifically, this last position claims that concepts are abstracted away from sensorimotor experience and organized according to conceptual properties. In particular, action verbs may reflect the retrieval of non-sensory, conceptual or grammatical information relevant to action verbs. Present results showed that different action verbs (i.e., to *grasp* and to *look at*) are specifically associated to different adverbs of space (i.e., *near* and *far*) depending on the possibility to perform that action at that space. That is, grasping is only possible in the near space. Therefore, these results are in favor of the first hypothesis and can hardly be explained by the second one.

While neuroscience research has devoted many resources to studying the role of the sensorimotor system in coding of physical space, in the era of smartphones, almost no studies are present on the influence that the acquisition of digital skills and the constant utilization of technological devices has on sensorimotor abilities and processes. The only few studies suggest that something interesting is happening there. For instance, touchscreen users showed larger amplitude of cortical potentials in response to tactile stimulation of the fingertips compared to nonusers, and amplitude was directly proportional to the recent phone use history [66]. Moreover, patterns of inter-touch intervals were different for content consumption and content generation, and the presence of recent greater activity in content generation influenced somatosensory cortical activity [26]. This suggests that digital actions with different functional characteristics can involve the sensorimotor system differently, just like actions performed in the physical world. Present results are in favor of this possibility, indicating that the encoding of the representation of digital space depends on the actions functions to be performed when using that specific app. Therefore, it is plausible to expect for the digital space the same consequences in terms of behavior, cognition, way of interaction, and pathologies found for the physical space [56]. For example, it is possible to assume that lesions to different cortical areas may induce neglect-like symptoms which are selective for content generation or content consumption. However, currently, no neuropsychological testing specifically considers digital skills among its trials, even if most neuropsychological assessments include at least one measure that is administered, scored, or interpreted by computers or touchscreen apps [67].

Obviously, this being the first study that deals with the relationship between spatial concepts and different apps, the absence of control conditions may limit the internal validity of the study. Further studies will be needed to rule out that the results do not depend on other factors such as, for example, different attitudes towards different apps. Among others, one possibility could be that participants might perceive WhatsApp as a more trustworthy app than Google. In this case, the *near*/*far* association could reflect a figurative closeness rather than a spatial or agency-related effect. We trust that in the future all alternative explanations will be taken into consideration allowing to verify the correctness of our interpretation.

A further and important limitation of the present study is evident, due to the restrictions caused by the COVID-19 pandemic which made it impossible to expand the experimental sample to other categories. All participants in the study were students belonging to generation Z (i.e., people born between 1996 and 2015). The average Gen Z received their first mobile phone at age 10.3 years, grew up playing with their parents’ mobile phones in a hyper-connected world, and the smartphone is their preferred method of communication. Together with Millennials (born between 1980 to 1996), they are defined as digital natives, as opposed to digital immigrants (i.e., Baby Boomers, 1946–1964, and Gen x, 1965–1980) [68]. Indeed, a characteristic of worldwide immigrants is their accent. For this reason, second-language learners are readily identified as non-natives, i.e., immigrants. This happens because their phonetic repertoire depends on their native language. The same can be said for the digital immigrants, who face the digital environment for the first time while possessing a cognitive and sensorimotor system totally forged by the continuous and exclusive interaction with the physical environment. This can determine a sensorimotor and cognitive “accent” that makes their spatial cognition of the digital environment different from the digital natives. The latter, indeed, can start interacting with the physical and digital environment almost at the same time. To verify the presence of differences in digital natives and immigrants, further studies are needed considering the generational cohort as a covariate.

## 7. Conclusions

Present findings suggest that the distinction in the use of proximal or distal space adverbs depends on the characteristics of the actions potentially suitable to be performed in that space. The results showed, for the first time, an implicit association between the adverbs of *near* and *far* and, respectively, actions that determine an effect in space (i.e., *grasp*) and actions that only allow the perception of objects in space (i.e., *look at*). This same result was also found for the digital space, as even in this environment our behaviors are divided into acting and perceiving. Specifically, results indicated an implicit association between the adverb *near* and app icons (i.e., *WhatsApp*) that direct to content generation actions and the adverb *far* and app icons (i.e., *Google*) that direct to content consumption actions. For the first time, present findings suggest that adverbs of space also apply to digital space.

## Figures and Tables

**Figure 1 brainsci-11-01523-f001:**
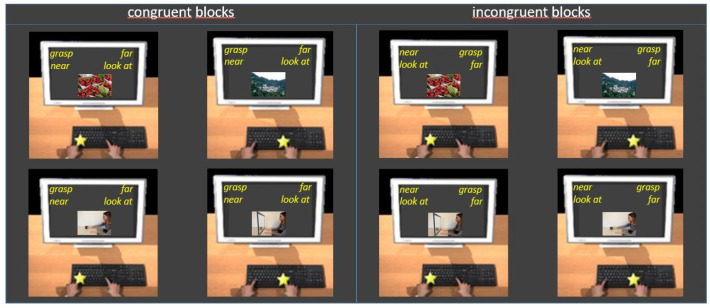
Examples of trials in congruent (**left panel**) and incongruent (**right panel**) blocks of Experiment 1. Participants were asked to press the E key with their left hand, when the category to which the image belongs was written at the top left, and to press the I key with their right hand, when it was written at the top right (see the text for details).

**Figure 2 brainsci-11-01523-f002:**
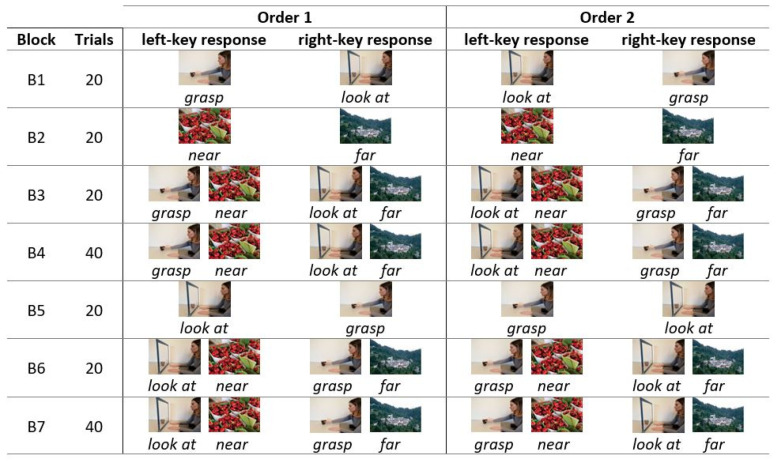
Structure of the seven blocks in the IAT administered to the two groups of participants (Order 1, Order 2) in Experiment 1 (see text for details).

**Figure 3 brainsci-11-01523-f003:**
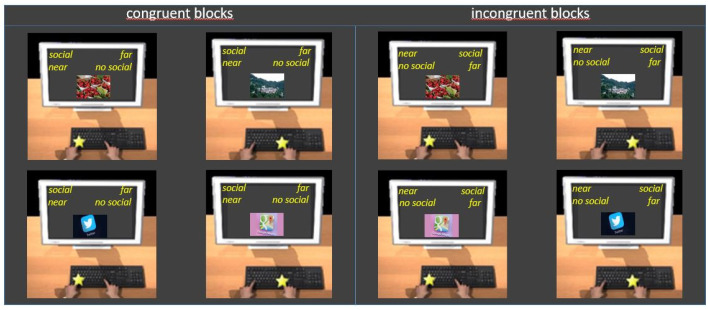
Examples of trials in congruent (**left panel**) and incongruent (**right panel**) blocks of Experiment 2. Participants had the instruction to press the E key with their left hand when the category to which the image belongs was written at the top left and to press the I key with their right hand when it was written at the top right (see the text for details).

**Figure 4 brainsci-11-01523-f004:**
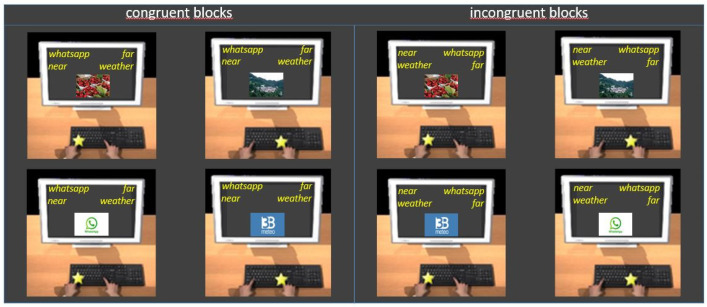
Examples of trials in congruent (**left panel**) and incongruent (**right panel**) blocks of Experiment 3. Participants were asked to press the E key with their left hand when the category to which the image belongs was written at the top left and to press the I key with their right hand when it was written at the top right (see the text for details).

**Figure 5 brainsci-11-01523-f005:**
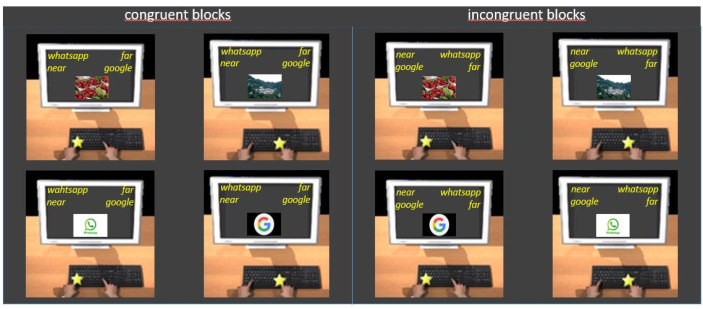
Examples of trials in congruent (**left panel**) and incongruent (**right panel**) blocks of Experiment 4. Participants were asked to press the E key with their left hand when the category to which the image belongs was written at the top left and to press the I key with their right hand when it was written at the top right (see the text for details).

**Figure 6 brainsci-11-01523-f006:**
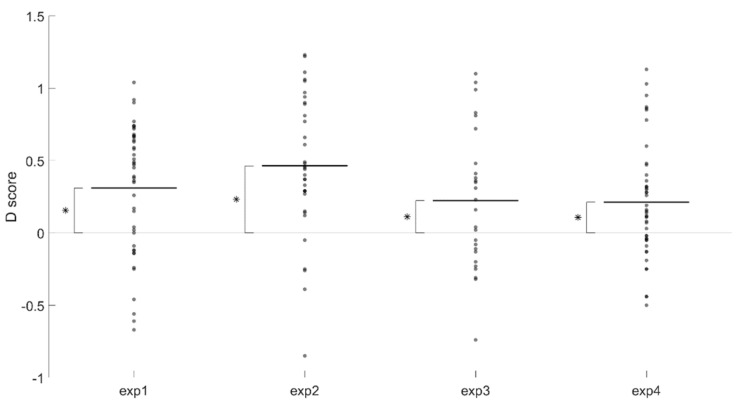
*D* scores of Experiments 1-4. Each marker indicates the *D* score for each participant. Short horizontal lines mark the mean *D* score for each experiment. Positive *D* scores indicate the presence of the predicted association. Asterisks show the presence of a statistically significant association, as a result of a one sample *t*-test of the mean *D* score against zero *D* score (horizontal line equivalent of no association).

## Data Availability

The data for each experiment are available at the link https://osf.io/9g235/?view_only=f072f3bb7ea34def93ea3cb07b9f2e78 (accessed on: 15 November 2021).

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
