# Peer review of "Implicit Associations between Adverbs of Place and Actions in the Physical and Digital Space"

_brainsci, 2021, doi:10.3390/brainsci11111523_

Round 1
Reviewer 1 Report
Craighero and Marini employ an implicit association paradigm, in which they measure the reaction time and error rate to categorise visual scenes. They investigate whether the reaction time and error rate are different when certain words are paired with the words describing the categories. They demonstrate that the adverb ‘near’ is more closely associated with words that imply the ability to alter the state of an object or online space (‘grasp’, ‘social’, ‘whatsapp’), while the adverb ‘far’ is more closely associated with words which do not alter the object or online space (‘look at’, ‘no social’, ‘weather’, ‘google’). The authors interpret these results as evidence that our use of spatial adverbs relies on a functional distinction between the effects of potential actions available in different parts of space, even in virtual spaces.
The manuscript is clear and well-written. The experimental methods and analyses are appropriate, and the four experiments synergistically support the conclusion. I only have minor comments and questions.
General coments:
1. Regarding the neural circuits involved in spatial processing.
I believe the authors could be a bit more careful in their statements about which areas underlie the observations described here. For example, VIP and F4 certainly seem to preferentially process the space near the body, but VIP is not unanimously assumed to be the hub of reaching and grasping actions. In fact, stimulation of F4 and VIP allegedly leads to more defensive/avoidance actions than appetitive actions (Graziano & Cooke 2006, Neuropsychologia). Rather, areas such as F2, MIP, and V6a, or F5, 7b, and AIP – which also seem to have an over-representation of events near the body – are more canonically described as being the major components in grasping and reaching circuits respectively (e.g. Cléry et al 2015, Neuropsychologia; Caminiti et al 2017, eNeuro; Serino 2019, Neurosci Behav Rev). On top of that, the ‘extrapersonal’ areas identified in the intro (LIP and FeF) have certain response properties which could be argued are distinctive of near-space representations (Cléry et al 2015, Neuropsychologia). Finally, it is still quite hotly debated whether ‘reaching space’ and ‘peripersonal space’ should be considered to be the same construct (Zanini et al 2021, Psychonimic Bull & Rev).
The above suggests that the authors should tone down the strength of their claims about specific areas (e.g. lines 88-91, 124-125, 297-299), and perhaps be clearer that the distinction between areas is not quite as clear cut as they make it seem.
2. Regarding the possibility to alter the object or online space:
This point is more speculative, but could be briefly addressed in the discussion. The authors conclude that the ability of a subject to act on an external entity is what differentiates near from far. In this framework, it seems like collecting information about an object without altering its state does not count as acting on an object (at least in the case of vision, e.g. lines 124-125, 180-181). However, we can easily imagine a physical object that can be explored by touch, but is so heavy and strong that its state cannot be meaningfully affected using the body alone. Under (what I understand to be) the authors’ framework, such a very strong and heavy object should never be considered ‘near’ – even if it is metrically very close to the body – because we can only gather (visuo-tactile) information on it, and cannot affect its state. This seems counterintuitive, but might be an experimental prediction worth testing in the future? E.g. a manipulable object should be more strongly associated with ‘near’ than an immovable one. Any (short) speculation on such a seeming paradox would be interesting.
3. Regarding online spaces and the correlation with social spaces:
In all experiments regarding virtual concepts in this paper, the ‘manipulable’ word/concept was a social one. Could the argument therefore be made that the association between social&near does not depend on manipulability, but rather the real-world statistics that to be social, one has to be close to others? The discussion of the paper could suggest a future experiment to test this in which the manipulable virtual space is a non-social environment, such as a single-player game, and the non-manipulable space is something like a movie. Of course if the authors would perform such an experiment, I feel their conclusion would be substantially strengthened, but I don’t believe it to be strictly necessary for publication.
Specific comments:
Line 128: Maybe the statement that perceiving and acting are binary behaviours is a bit strong
Line 273: should “(computed in step c)” be “(computed in step b)” instead?
Line 299: This also relates to point 1 above, but affordances to objects in the space near the body also include actions away from stimuli. In fact Graziano seems to believe this is the main purpose of the F4-VIP circuit (see his recent book ‘The Spaces Between Us').
Line 301: I believe this should be a different circuit
Lines469-470: I don’t understand what reference [26] has to do with this sentence (same for the mention of reference [26] in the introduction.
Reviewer 2 Report
In this manuscript, Craighero and Marini investigated the strength of the semantic association between spatial concepts (near vs far) and different agentive modalities (acting vs observing), in either physical or digital environments. The authors hypothesise – and convincingly show – that the concept of ‘near space’ is associated with functional actions, i.e. context which imply an interaction with the physical/digital target, like grasping an object or generating digital content. In contrast, the concept of ‘far space’ is associated with observational actions, i.e. context where an interaction is precluded, like looking at an unreachable object or consuming digital content.
The paper is well-written, the study appears to have been carefully performed, and the conclusion is sound. However, I have a few clarification questions and comments which I’d like the authors to address.
1. Did the authors collect any data on digital app usage from their participants? Many people might be using platforms such as Instagram or Twitter more for content consumption than for content generation. This point seems crucial for a clear operationalisation of functional vs observational actions in digital environments. Do the authors have a way to discriminate between these two types of behaviours in their participants? I don’t think this constitutes a major concern for the study, as a blurred boundary between content generation/consumption would count against the hypothesised association between adverbs of place and different apps. Yet, a comment would be helpful, I think.
2. Another concern is that people might have different attitudes toward different apps. For example, participants in experiment 4 might perceive WhatsApp as a more trustworthy app compared to Google, such that the near/far association could reflect a figurative closeness rather than a spatial or agency-related effect. Could the authors please clarify this point? In general, a control condition confirming that people that do not have any previous experience with the digital apps do not show a positive D score seems necessary, or at least advisable. The authors already mention this limitation in their discussion. However, I’d suggest reworking that paragraph to limit speculations on “sensorimotor and cognitive ‘accent’”, and focus instead on the how the absence of such a control condition limits the internal validity of their study.
3. The IAT tests the semantic closeness of two concepts. Therefore, it’s not immediately clear how well the (semantic) effect described by the authors can relate to the brain areas they mention. Those areas might be involved in the encoding of near/far space for functional/observational actions, but such encoding is not tested directly in the present study. A direct test (rather incompatible with IAT procedures) would require that participants actually take agentive vs observational roles. I suggest that the authors clarify or tune down their claims. Similarly, the first sentence of the general discussion is also misleading, as the authors did not test “the role of the somatosensory system” per se, but only the semantic association between adverbs of place and the *concept* of acting vs perceiving.
Minor points:
- The information about the structure of the IAT blocks could be more clearly presented as a table rather than text.
- The figures could read “congruent/incongruent blocks” instead of “combined blocks, adverb + action/reverse action”. Also, Figures 2-4 could be simplified, as the 4 x 2 panel depicting every condition of the crucial blocks in each experiment seems unnecessary.
- The number of participants is highly variable across the 4 experiments (26 < n > 44). How was the stopping rule for each experiment determined? Did the authors perform any a-priori power analysis to estimate their sample sizes?
- Please report effect sizes along with the statistical values of each t-test.
- Please use italic or quotation marks when mentioning ‘near’ and ‘far’ as adverbs of place e.g. ll. 146-151.
Round 2
Reviewer 2 Report
The Authors have adequately clarified all my points, and I can now confidently recommend publication.